# Pros and Cons of Cryopreserving Allogeneic Stem Cell Products

**DOI:** 10.3390/cells13060552

**Published:** 2024-03-21

**Authors:** Caterina Giovanna Valentini, Claudio Pellegrino, Luciana Teofili

**Affiliations:** 1Dipartimento di Diagnostica per Immagini, Radioterapia Oncologica ed Ematologia, Fondazione Policlinico Universitario “A. Gemelli” IRCCS, 00168 Rome, Italy; caterinagiovanna.valentini@policlinicogemelli.it (C.G.V.); claudiopellegrino94@gmail.com (C.P.); 2Sezione di Ematologia, Dipartimento di Scienze Radiologiche ed Ematologiche, Università Cattolica del Sacro Cuore, 00168 Rome, Italy

**Keywords:** COVID-19, allogeneic hematopoietic stem cell transplant, cryopreservation

## Abstract

The COVID-19 pandemic has precipitously changed the practice of transplanting fresh allografts. The safety measures adopted during the pandemic prompted the near-universal graft cryopreservation. However, the influence of cryopreserving allogeneic grafts on long-term transplant outcomes has emerged only in the most recent literature. In this review, the basic principles of cell cryopreservation are revised and the effects of cryopreservation on the different graft components are carefully reexamined. Finally, a literature revision on studies comparing transplant outcomes in patients receiving cryopreserved and fresh grafts is illustrated.

## 1. Introduction

The role of hematopoietic stem cell transplantation in the treatment of hematologic and non-hematologic malignancies is rapidly expanding. Current therapeutic protocols comprise cryopreservation of hematopoietic progenitor cells (HPCs) for virtually all autologous transplants, although successful experiences with fresh grafts have also been reported [1,2]. On the other hand, grafts for allogenic use have been typically employed fresh. One notable exception is represented by umbilical cord blood (UCB) [3]: the actual transplant is harvested at the time of birth and used at a later point in time for an often-indeterminate recipient, requiring cryopreservation in a cord blood bank for an indeterminate amount of time. Notably, UCB units stored over 29 years of cryopreservation still exhibit a high quality in terms of viability of total nucleated cells and CD34+ cells [4]. In 2020, the COVID-19 pandemic has precipitously changed the practice of transplanting fresh allografts. The lockdown disrupted international and domestic travels, into the hospitals new paths dedicated to COVID-19 patients replaced pre-existent units and to access any hospital departments become much more problematic. The complex logistics of the allogenic transplant in which graft collection was accomplished to be promptly delivered on the day of planned transplantation, was seriously threatened. In addition, there was a realistic risk that the donor could became SARS-CoV-2-infected and then unable to donate after the recipient had initiated the conditioning. To circumvent the potential critical impact on allogeneic transplant candidate patients, scientific societies, donor registries, and competent authorities worldwide recommended cryopreserving and securing donations prior to initiation of recipient conditioning [5,6].

Before the pandemic, graft cryopreservation was unusual and was mostly performed for related donors, whilst only 5–8% of unrelated transplants were cryopreserved [7]. This review aims to illustrate how the last three years have changed the current knowledge and perception of the impact of cryopreservation on transplant outcomes.

## 2. Basic Principles of Cryopreservation of Cells and Tissues

Cryopreservation refers to the technique of storing biological materials at below-zero temperatures, slowing the rate of degradation to ensure minimal loss in function; this practice has wide-reaching applications, including basic biological research, agriculture and food industry, and medicine. Biological materials experience a significant decrease in kinetic energy and molecular motion when exposed to ultralow temperatures below −130 °C. This slowdown causes a decline in the rates of both chemical and biological reactions. Consequently, basic cellular processes, such as metabolism, active transport, enzymatic reactions, and diffusion, also slow down. As a result, the material can remain suspended until the temperature is increased again [8].

Although ultra-low temperatures themselves do not directly cause physical damage, damage can occur during the process of freezing and subsequent thawing. When cooling aqueous solutions below the freezing point, ice crystals will form in the extracellular media. This will decrease the concentration of extracellular water in the sample, while solutes that were previously distributed in the bulk solution will become concentrated in the residual water channels between the ice crystals. Cells that are enclosed in these channels will experience an increasing solute concentration than in the sample solution without ice present. This, in turn, creates an osmotic gradient across the cell membrane, causing cellular dehydration, which ultimately leads to osmotic shock and increased toxicity.

In addition, excessive intracellular ice formation determines irreversible damage to cell membranes and intracellular organelles. In most scenarios, an intermediate cooling rate during freezing maximizes cell survival, allowing cells to gradually dehydrate and avoid excessive osmotic stress and intracellular ice formation [9]. An alternative approach to preserve biological samples is to use extremely high cooling rates to bypass the crystalline (ice) phase and obtain an ultrahigh-viscosity amorphous glass state, which is known as vitrification. This method helps to prevent cellular damage caused by ice nucleation and intracellular ice growth, as during supercooling, ice nuclei lack sufficient time to grow due to water molecule diffusion limitations. However, vitrification requires high cryoprotectant (CPA) concentrations to increase medium viscosity resulting in osmotic stress during CPA loading and removal. Moreover, achieving homogenous rapid freezing of large sample volumes is technically challenging, limiting the application of these protocols to cellular therapies. On the other side, vitrification is the clinically preferred method to cryopreserve oocytes and embryos, with reports of better clinical outcomes compared with slow-freezing protocols [10]. Recently, Akiyama et al. demonstrated the feasibility of directly vitrifying mammalian cells in a CPA-free medium by ultrarapid cooling using inkjet cell printing, a technique named super flash freezing, thus minimizing osmotic toxicity [11].

During the thawing process, dehydrated cells can face exposure to non-physiological volumes of water or buffer solutions, which can result in swelling and cell lysis, while preformed ice crystals can undergo recrystallization, where larger crystals grow at the expense of smaller ones, adding mechanical damage. To achieve a successful rewarming, both uniform and fast heating rates are required. However, external heating, the most common applied thawing method, causes the onset of thermal gradients as the outer part of the sample melts faster than the inner part. On the contrary, nanowarming exploits the local heating effect associated with magnetic nanoparticles exposed to an alternating magnetic field, allowing homogeneous and rapid rewarming of biological samples including human induced pluripotent stem cells [12,13].

Some of these cryopreservation challenges can be partially addressed by the addition of cryoprotectants [14,15]. Dimethyl sulfoxide (DMSO) and glycerol are examples of permeating cryoprotectants that can easily enter the cells, while non-permeating extracellular cryoprotectants are made up of macromolecules such as hydroxyethyl starch (HES) and small molecules like trehalose. The specific ways in which individual cryoprotectants act, having not been fully clarified yet. It is generally believed that most cryoprotectants have multiple modes of action such as modulation of the hydrogen bonding and the properties of the cell membrane, dilute solute effects, and increase in the solution viscosity at low temperatures [15]. 

There has been extensive research on formulating and improving cryopreservation media to achieve the best possible cellular outcomes. Furthermore, there has been a growing interest in the characterization of innovative agents to prevent biophysical damage caused by ice growth, such as ice-binding proteins, nucleation modulators, ice recrystallization inhibitors, new macromolecular cryoprotectants, and vitrification agents [8].

In addition, the multiple stress factors of the freeze–thaw process trigger a complex molecular biological stress response, culminating in the activation of apoptotic and secondary necrotic processes, ultimately leading to cell death within hours or days after thawing, a phenomenon termed cryopreservation-induced delayed-onset cell death (CIDOCD) [16]. This discovery resulted in a paradigm shift in the cryopreservation sciences from a primarily chemo-osmometric (ice control) approach to an integrated one also combining molecular modulation of cellular pathways to minimize CIDOCD [17]. Unsurprisingly, targeting apoptotic caspase activation, oxidative stress, unfolded protein response, and free radical damage in the initial 24 h post-thaw, resulted in increased overall cell survival of human hematopoietic progenitor cells [18]. In addition to cryopreservation-related variables, the extent and timing of CIDOCD varies also according to different cell populations as biochemical pathways dysregulated after freezing and thawing may be cell specific. As an example, the post-thaw application of Rho-associated protein kinases inhibitors to T-cell cultures reduces the membrane expression of Fas death receptor, increasing cryopreserved cell yield [19].

It is important to recognize that cryopreservation can alter cell phenotype and function following thawing, including changes in surface markers levels [20] and long-term gene expression [21]. Consequently, an increasing number of assays have been developed to analyze the post-thawing cell quality, encompassing assessment of membrane integrity, molecular mechanisms, cell function, and biochemical alterations [22]. 

## 3. Cryopreservation of Hematopoietic Stem Cell Grafts

Most evidence regarding the biology of cryopreservation of HPC grafts derives from studies on autologous and cord blood settings. No individual cryopreservation method has been universally adopted; procedures may vary across transplant centers [23]. In current practice, guidelines for HSC cryopreservation suggest 5–10% dimethyl sulfoxide (DMSO) as a cryoprotectant, freezing at a controlled rate of 1–2 °C per minute and storage in vapor phase nitrogen at a temperature of ≤−140 °C. Cryopreserved bags must be thawed on site of transplantation and reinfused within 10–20 min using a standard transfusion filter [24].

Typically, a controlled rate freezing approach is used in most protocols, wherein the concentrated HPCs are frozen at a speed of 1–2 °C per minute until they reach a temperature point of approximately −40 °C. After that, the freezing process is accelerated to a pace of roughly 3–5 °C per minute to reach the target temperature. Uncontrolled rate freezing techniques are also feasible and represent an attractive alternative, being less time-consuming and requiring less high-level technical expertise [25,26], but hematologic reconstitution may be suboptimal [27]. The best storage conditions for optimal preservation are in the vapor nitrogen phase at a temperature of −156 °C. Thawing of cryopreserved HPC using dry warming or a 37 °C water bath gives similar viability, apoptosis/necrosis rate, and clonogenic potential: nevertheless, the dry warming procedure ensures a lower degree of microbial contamination [28].

DMSO is the current gold standard for cell cryopreservation and is the most used intracellular cryoprotectant for HPCs [23]: it is usually diluted by human albumin, plasma, or another solution licensed for clinical application to reach a final concentration of 10% [29]. Nevertheless, the DMSO efficacy needs to be balanced with its pleiotropic effects exerted on cell biology and apparent toxicity in recipient patients [30]. In fact, DMSO infusion can determine clinically significant side effects in recipients [31]. Most common adverse reactions reported involve gastrointestinal (nausea, vomiting, diarrhea), cardiovascular (hypotension, bradycardia) and respiratory system (dyspnea, lung edema) and skin (erythema, pruritus, rash) [32,33]. Most of these events may be ascribable to the DMSO-induced histamine release [34]. A correlation between the total DMSO dose infused and the total number of side effects has also been described [35]. Several approaches have been suggested to minimize the likelihood of toxicity associated with DMSO. The most intuitive approach is to remove the DMSO from the graft after thawing, either by simple centrifugation or by more advanced techniques, including spinning membranes and microfluidic channels [33]. 

It has been shown that DMSO depletion reduces the frequency of adverse effects, without significant loss of CD34+ cells, viability, and colony-forming unit-granulocyte-macrophage activity [36,37]. Unfortunately, this procedure may be associated with delayed platelet engraftment [36]. Furthermore, the washing is time-consuming and may cause cell clumping or contamination of the product. For this reason, most transplant centers avoid routinely carrying out this procedure [29]. Another strategy to minimize the final volume and absolute amount of DMSO is concentrating the cells before cryopreservation. However, once again, there may be a risk of cell clumping after thawing [38]. Finally, reducing the concentration of DMSO in the cryoprotective mixture to 5% effectively minimizes the amount of infused DMSO, resulting in a lower incidence of adverse events, without affecting hematopoietic recovery [39]. In the next future, new DMSO-free freezing media could replace DMSO and provide adequate cryoprotection to HSC grafts limiting toxicities to cells and recipients [40].

Considering that prolonged exposures to high concentrations of DMSO are toxic and that even low concentrations can elicit profound epigenetic modifications in some cell types [41], concerns have been raised about the effects of DMSO on HPCs [42]. However, DMSO seems not to be toxic after short-term exposure at the concentrations used for cryopreservation of bone marrow (BM) and peripheral blood stem cells (PBSCs), as the clonogenic potential is preserved [43,44]. After cryopreservation, neither recovery nor viability of CD34+ cells from both HPC-leukapheresis products and CB are functions of the concentration of DMSO [45]. The expression of adhesion molecules is not appreciably changed by freeze–thaw and DMSO exposure, except for a mild downregulation of CD62L (L-selectin), which is rapidly reversed in vivo after the transplantation [46]. In contrast, short priming with DMSO before transplantation may favor HPCs differentiation [47], chemotaxis, and homing to the bone marrow niche [48], ultimately favoring the engraftment.

CD34+ cell recovery and viability are variable after thawing cryopreserved HPC products. It should be preliminary noted that there is a considerable degree of variability between different laboratories concerning the methods employed for testing the viability of HPCs. Various techniques such as flow cytometry [49] and image-based assays have been used for testing both before and after the cryopreservation process [22]. It is noteworthy that there is currently no standardization of testing methods, which may have implications for the establishment of universal viability thresholds [50].

Despite these limitations, the post-thaw viable CD34+ cell count may represent a good predictor of in vivo hematopoietic engraftment and a good estimator of the quality of the product issued, both in autologous [51] and allogenic settings [52]. 

Most cryopreserved allogenic stem cell products are usually infused within months from harvesting, minimizing the possible additional detrimental effect of long storage on CD34+ quality. Moreover, most of the loss of CD34+ cells tend to occur during pre-storage manipulation and subsequent thawing, as storage for up to 10 years does not significantly affect the number, viability, metabolic function, and clonogenic potential of hematopoietic stem cells [53]. On the other side, for allogenic products, a significant amount of time may elapse between donation and cryopreservation. The total time can be further sub-classified as a “transit time” (time elapsed between the end of collection at the donation site and the delivery to the recipient center), “holding time” (time between receipt and the actual beginning of processing), and “processing time” (total time required to completely process and store the product). These variables need to be considered, particularly when dealing with products of international origin. Maurer et al. reported that use of cryopreserved products with a total processing time >48 h trends toward lower day 30 CD3+ donor chimerism and an increased risk of graft failure [54]. By way of analogy, reduced post-thaw CD34+ cell recovery and viability of allogeneic HPC products are linked to a longer storage time before cryopreservation [2,55], while minimizing the transit time below 36 h prevents poor viable CD34+, CD3+ cells, and CFU-GM recoveries [56]. In contrast, Reddy et al. described that extended transport may contribute to lower post-thaw viabilities of total nucleated cells, but without affecting CD34+ cell recoveries [57]. Leukocytapheresis variables also affect the quality of the cryopreserved products: higher volumes, concentrations of total nucleated cells, white cell counts, and hematocrit at freezing all decrease the final CD34 viability [2,55,56,57,58]. These observations may help to optimize leukocytapheresis practices when collecting allogenic HPCs not to be used fresh. 

The cell compositions of HPC-enriched grafts is highly complex, comprising not only CD34+ cells, but also lymphocytes (including B cells and T cell subsets such as regulatory T cells and cytotoxic T cells), natural killer (NK) cells, and other subsets of innate lymphoid cells (ILCs), dendritic cells (DC), and myeloid-derived suppressor cells (MDSC). 

Graft composition not only contributes to hematopoietic and immune reconstitution but is also associated with transplant outcomes, including graft-versus-host disease (GvHD), relapse, transplant-related mortality (TRM), and overall survival (OS) [59,60]. This variety depends in first place to donor related factors, source of the graft (bone marrow, mobilized peripheral blood, cord blood) and harvest modality. However, when cryopreserving heterogeneous cell samples, such as BM or PBSC grafts, the cryopreservation process itself can differentially select for cellular sub-populations which may be more tolerant to the molecular and physical stresses experienced during the freeze–thaw: to this end, cryopreservation can be rightfully considered an involuntary graft manipulation process. 

As an example, the different subpopulations of human peripheral blood mononuclear cells display unique sensitivities to cryo-storage [61]. By way of analogy, in unmanipulated leukapheresis products from G-CSF–mobilized donors, CD34+ and CD19+ cells show greater tolerance for cryopreservation and thawing compared with CD3+ cells [62]. In fact, among immune cells, T cells seem vulnerable to damage induced by cryopreservation, affecting their viability, functionality, and suitability for immune assays [63,64,65]. 

The two most common T cell-based therapies that usually undergo cryopreservation are Chimeric Antigen Receptor T cell (CAR-T) and Donor Lymphocyte Infusion (DLI). The effective manufacturing of CAR T cell products relies on smooth coordination of the supply chain post-collection: in this setting, cryopreservation ensures flexibility for the timing of leukapheresis and overall simplifies the logistics of the whole process. Current guidelines recommend cryopreservation of leukapheresis products on the same day as collection to improve post-thaw cell viability and manufacturing outcomes. Freezing seems not to significantly impact the function of the final product: except for a decrease in cytokine release, the cryopreserved CAR-T cells retain their antitumor functions [66], in vivo levels, persistence, and clinical response [67].

Donor T cells are used in adoptive immunotherapy after allogeneic transplantation, either for therapeutic purposes in cases of disease relapse or progression or as a prophylactic measure in patients at high risk of relapse. DLIs are often collected and processed around the time of HSC donation and then cryopreserved in 10% DMSO and stored in the vapor phase of liquid nitrogen until needed. DLI are administered gradually increasing the cell dosages to minimize the risk of GvHD [68].

Overall, the scientific consensus on the effect of cryopreservation on T cell functions remains divergent, in large part due to the heterogeneous nature of the samples being sourced for analysis. To further complicate the picture, each T-subset responds differently to cryopreservation. Freezing produces a significant decline in the rates of naive and “central memory” T cells, paralleled by an increase in “effector” CD8+ T cells [69]. Helper T cells show higher post-thaw recovery than cytotoxic T cells in both DMSO and DMSO-free medium [61], while NK cells are known to suffer from poor cryopreservation outcomes [70]. Cryopreservation may have a detrimental effect on Tregs [71], can decrease their viability, cause abnormal cytokine secretion, and compromise the expression of surface markers essential for proper Treg function and processing [72], limiting their clinical therapeutic applications.

NK cells are particularly prone to freeze–thaw and DMSO-induced damage [73,74], leading to a distinct, functionally compromised CD56 dim CD16 negative phenotype [75]. Cryopreservation with DMSO was also shown to lead to a reduced expression of tumor necrosis factor (TNF)-related apoptosis-inducing ligand (TRAIL) and natural killer group 2D (NKG2D) on NK cells, alongside reduced NK cell cytotoxicity [76]. Moreover, cell migration is also impaired [77].

On the contrary, the ability of dendritic cells to generate antigen-specific reactions is preserved once thawed [78]: this is of particular interest, since the number of plasmacytoid dendritic cells in allogenic graft has been associated with increased overall survival and fewer deaths resulting from GvHD or from graft failure [79].

These considerations underscore the difficulty of optimizing the viability and functionality of all cellular subsets in a heterogeneous population with a universal cryopreservation-thawing protocol. Recently, there has been growing interesting in developing robust DMSO-free cryopreservation methods that can improve product safety while maintaining cellular viability and efficacy [80]: this approach has been proven feasible for both T cells [81] and NK cells [82]. A more conservative approach consists of the reduction in DMSO. In fact, 10% DMSO results in reduced viability of different CD4-positive T cell populations, including T-helper, T-cytotoxic, and T-regulatory populations, and a decrease in their proliferative and cytotoxic response to immunologically relevant stimuli. Conversely, using solutions containing 5% DMSO with intracellular-like cryoprotectant stabilizers could instead maintain T cell function at levels similar to refrigerated control samples [63].

## 4. Clinical Impact of Allogenic Peripheral Blood Stem Cell Cryopreservation

We carried out a systematic search strategy using the PubMed database to identify studies comparing the outcomes of allogeneic hematopoietic stem cell transplant using cryopreserved or fresh grafts. The following queries were utilized: All fields: [(cryopreserved) AND (transplant) AND (hematopoietic)]. Search results were managed using the Rayyan application [83]. We excluded duplicates, papers not suiting the searched topic, those not including original data, communications at congresses, and papers not in English. C.P., C.G.V., and L.T. independently controlled all references, and discrepancies were discussed and resolved together. In total, 1739 references were identified on 15 January 2024. In the end, 29 papers were included [54,55,84,85,86,87,88,89,90,91,92,93,94,95,96,97,98,99,100,101,102,103,104,105,106,107,108,109,110] (Figure 1). 

These studies investigated the effect of cryopreservation on short- and/or long-term transplant outcomes according to a retrospective design. Basically, they could be separated in two different groups. The first group (Table 1) includes studies carried out before or immediately after the onset of the pandemic, in which the reason for cryopreserving was not related to the COVID-19 spread. Basically, in these cases, graft cryopreservation often ensued to unexpected complications in the clinical course of recipients, potentially affecting the transplant outcomes. In contrast, Table 2 includes studies in which graft cryopreservation was the common practice in the pandemic period, independently from specific recipient clinical conditions. 

In Figure 2, we summarized the findings of all studies included in our revision regarding neutrophil and platelet engraftment. The effects of cryopreservation on acute and chronic GVHD and survival and relapse rates are recapitulated in Table 1 and Table 2. 

The original experience on cryopreserved allogenic grafts was mainly based on bone marrow [84]. In 2006, Frey et al. reviewed literature data on cryopreserved allogenic grafts and found that information was limited to small case series and retrospective cohort studies from individual institutions [112]. In total, data related to 67 transplants of cryopreserved stem cell products were identified, 57 from related donors and 10 from unrelated donors. In all cases, bone marrow was the stem cell source, except from a single case report using a cryopreserved peripheral blood product. There was no difference in significant clinical outcomes such as time to platelet and neutrophil engraftment or day 100 survival when compared with transplants using fresh grafts. Interestingly, in one of the studies included in this review, a statistically significant reduction in the incidence of acute GvHD was observed for 10 recipients of cryopreserved BM compared with 33 unmatched institutional controls [85]. Nevertheless, cryo- and fresh-graft recipient groups significantly differed for age, disease type, disease stage, and type of GvHD prophylaxis and accounting for these differences in a statistical regression model failed to identify cryopreservation as an independent predictor of GvHD [85]. As emphasized by Frey et al., the common approach of using fresh allografts derived from various theoretical concerns. First, cryopreserving grafts could result in the loss of a part of progenitor cells. Overall, freezing could selectively damage specific cell populations, in particular T lymphocytes, thus affecting the antitumor effect of the allograft Moreover, cryopreservation, freezing, and thawing, all require more extensive manipulation, with a higher risk for microbial contamination. Finally, the DMSO-related toxicity adds further danger to the transplant procedure [112]. The authors concluded that the effect of freezing on graft cell subsets deserved to be better investigated, considering that cryopreserving allografts could effectively simplify the demanding logistics of related and unrelated transplants [112]. 

The subsequent studies, whose principal findings are summarized in Table 1, mainly focused on PBSC grafts.
cells-13-00552-t001_Table 1Table 1Impact of cryopreservation in studies including patients transplanted before COVID-19 pandemic.Authors [Ref]Study TypeDisease/Donor TypesPatients (Graft Source) *Controls (Graft Source) **aGvHD Cumulative Incidence (%)cGvHDCumulative Incidence (%)Overall Survival (%)Relapse Rate (%)Other FindingsKim DH et al., 2007 [86]Single centerMalignantdiseases/related donors105(PBSCs)106(PBSCs)Grade II–IVCryo 78.2 ± 4.3Fresh 81.2 ± 4.51 year Cryo 83.8 ± 5.1Fresh 90.6 ± 3.41 yearCryo 64.3 ± 5.1Fresh 65.1 ± 4.62 year Cryo 52.7 ± 6.5Fresh 59.4 ± 4.82 years Cryo 26.6 ± 5.8Fresh 19.4 ± 4.3Lymphocyte recovery >0.5 × 10^9^/LCryo 22 days Fresh 22 days >1 × 10^9^/LCryo 33 days Fresh 33 days % 1 year NRM Cryo 24.6 ± 4.6Fresh 20.4 ± 4.2Lioznov M et al., 2008 [87]Single centerNR/related and unrelated donors39 (31 PBSCs,8 BM)493 (PBSCs)NRNRNRNR% GF in PBSCs Cryo 19Fresh 1.4Medd P et al., 2013 [88]MulticenterMalignantdiseases/related and unrelated donors76(PBSCs)123 (PBSCs)Day +100 Grade II–IVCryo 31.7Fresh 36.91 year ExtensiveCryo 40.3Fresh 28.32 yearsCryo 45.3Fresh 60.32 years Cryo 35.3Fresh 36.9% 1-year TRMCryo 14.6Fresh 17.9% 2 year RFS Cryo 41.9Fresh 51.2Parody R et al., 2013 [89]MulticenterMalignantdiseases/matched related donors 224 (PBSCs)107(PBSCs)Day +100 Cryo 61.5Fresh 44% *p* < 0.001Grade II–IVCryo 44 Fresh 303 years extensive Cryo 50Fresh 423 yearsCryo 58Fresh 46Cryo 35%Fresh 40%% Day +100 NRM Cryo 15Fresh 9% 1-year NRM Cryo 24Fresh 16Eapen M et al., 2020 [90]Multicenter Aplastic anemia/related and unrelated donors52 (19 PBSCs, 33 BM)195 (63 PBSCs,132 BM) Day +100Cryo 12 Fresh 13 1 yearCryo 23Fresh 28 1 yearCryo 73 Fresh 91*p* = 0.0008Confirmed in PBSCs but not BMNR% 1-year GFCryo 19 Fresh 10*p* < 0.001 Confirmed in PBSCs but not BM Hamadani M et al., 2020[91]Multicenter, CIBMTR database/propensity score matched Malignantdiseases with ptCy/related HLA- or haploidentical, and unrelated donors 274(256 PBSCs,18 BM)1080(1009 PBSCs,71 BM)Day +100 Grade II–IVCryo 34 Fresh 31.3 1 yearCryo 26.8 Fresh 30.7 2 years Cryo 58.7 Fresh 60.6 *p* = 0.04 at regression analysis2 yearsrelapse/progression rateCryo 36.3 Fresh 30.7% 2-year NRMCryo 22.0Fresh 19.0 % 2-year DFSCryo 41.7 Fresh 50.4*p* = 0.03Hsu JW et al., 2021[92]Multicenter, CIBMTR databaseMalignantdiseases/related and unrelated donors7397(1051 related PBSCs; 678 unrelated PBSCs; 154 BM)5514(3030 related PBSCs; 2028 unrelated PBSCs; 456BM)Day +100 Grade II–IVRelated PBSCsCryo 35Fresh 30*p* = 0.01Unrelated PBSCsCryo 39Fresh 40BMCryo 31Fresh 33NRNo difference forBM andrelated PBSCsUnrelated PBSCs Cryo 57Fresh 46 *p* < 0.0012 years BMCryo 31Fresh 25Related PBSCsCryo 30Fresh 31Unelated PBSCsCryo 28Fresh 25In multivariate analysisLower aGVHD in related cryo PBSCssimilar TRM, OS, and PFS between cryo- and fresh grafts in BM and related PBSCs.In unrelated PBSC multivariate analysis confirmed lower OS (*p* < 0.001), lower PFS (*p* < 0.001), higher TRM (*p* < 0.001), and higher relapse rate (*p* = 0.002) Dagdas S et al., 2020[93]Single-center Malignantdiseases/full-match sibling donors 30 (PBSCs)42(PBSCs)Cryo 33.3Fresh 28.6Cryo 40Fresh 38.1Less liver cGvHD in cryo*p* = 0.0461 year Cryo 59Fresh 603 years Cryo 54Fresh 57Cryo 30Fresh 16.7% Day +100 NRMCryo 7 Fresh 19% 1-year NRM Cryo 13Fresh 22% 3-year NRM Cryo 26Fresh 30* Patients recipient of cryopreserved grafts; ** patients recipient of fresh allografts. OS, overall survival; RR, relapse rate; TRM, transplant-related mortality; NRM, non-relapse mortality; PFS, progression-free survival; aGvHD, acute graft-versus-host disease; cGvHD, chronic graft-versus-host disease; PBSCs, peripheral blood stem cells; BM, bone marrow; NR, not reported; ptCy, post-transplant cyclophosphamide; GF, graft failure; RFS, relapse-free survival.


In 2007, Kim et al. compared transplant outcomes in 105 patients receiving cryopreserved PBSC allografts from related donors to those of a historic control of 106 patients transplanted with freshly procured PBSCs. The median length of cryopreservation between collection and thawing was 15 days (range: 5–238 days). The authors did not report severe adverse events related to DMSO toxicity, with mild nausea/vomiting occurring mostly in female recipients, while bradycardia and hypotension were dependent on the total amount of DMSO infused. No microbial contamination of cryopreserved grafts was observed. As compared to controls, patients receiving cryo-allografts had similar engraftment kinetics of neutrophils, platelets, and lymphocytes. Among six patients not achieving engraftment, one case received cryopreserved PBSCTs and five fresh PBSCTs. The cumulative incidence of acute GVHD (grades II-IV) was 78.2% ± 4.3% after cryopreserved and 81.2 ± 4.5% after fresh PBSCTs (*p* = 0.113). The same figures for 1-year chronic GVHD were 83.8 ± 5.1% and 90.6 ± 3.4% (*p* = 0.673). Similar rates of 1-year and 2-year overall survival (OS), non-relapse mortality (NRM), and relapse were observed. In particular, the 2-year relapse rate was 26.6 ± 5.8% for patients receiving cryopreserved allografts and 19.4 ± 4.3% for those receiving fresh grafts (*p* = 0.340). Selective damage on megakaryocytic progenitor cells was reported. In the multivariate analysis, cryopreservation did not influence OS, NRM, and disease recurrence. The authors concluded that cryopreservation does not seem to alter graft function including the graft-versus-malignancy effect [86]. In 2008, Lioznov et al. retrospectively analyzed 31 frozen allogeneic PBSCs and 8 bone marrow grafts by flow cytometry regarding their CD34+ content, membrane integrity by 7-aminoactinomycin D (7-AAD) staining, and stem cell-specific enzyme activity (aldehyde dehydrogenase, ALDH) in relation to individual transplantation outcomes [87]. The authors found that 7-AAD-positive CD34+ cells significantly increased in cryopreserved PBSCs but not in BM compared to fresh allografts. Overall, 9 out of 33 patients (27%) who received unrelated cryopreserved PBSC allografts did not achieve engraftment, in comparison with 7 out of 493 recipients (1.4%) of fresh allogeneic PBSC grafts. CD34+ cells from PBSCs, but not bone marrow grafts, also showed a significantly reduction in ALDH content. The damaging effect of freezing in PBSC but not BM grafts could be in part ascribed to the higher content of granulocyte contamination [57]. Nevertheless, since PBSC allografts were all from unrelated donors and had been transported from collection site to the transplant center before being frozen, the authors suggested that PBSC grafts become much more sensitive to cryopreservation after transport and/or storage [87]. In 2013, Medd et al. compared 76 cryopreserved PBSC transplants from related (*n* = 57) and unrelated donors (*n* = 19) to a series of 123 fresh PBSC transplants [88]. The authors found a significant delay in neutrophil and platelet engraftment in cryopreserved transplants (HR 1.44 for neutrophil recovery, *p* = 0.003; and HR = 1.85 for platelet recovery, *p* < 0.001). No significant differences were observed regarding the incidence of acute and chronic GVHD and overall survival, whereas a slight but not significant higher rate of extensive chronic GVHD was observed among patients receiving cryopreserved grafts [89]. In terms of variance, Parody et al., comparing fresh (*n* 107) or previously frozen PBSC (*n* 224) transplants, reported a faster neutrophil recovery in the cryo-group [89].

Following the recommendations issued at the onset of the pandemic, several studies explored the impact of cryopreservation in retrospective series of patients receiving cryopreserved allografts before pandemic. Different transplant settings were investigated. Eapen et al. examined the effect of cryopreservation of related and unrelated donor grafts in transplantation for severe aplastic anemia [90]. The authors compared 52 recipients of cryopreserved grafts to 194 recipients of fresh grafts transplanted in the United States during the period from 2013 to 2019. Groups were matched for age, donor type, and graft type (PBSC or BM). There was no difference between groups regarding the hematopoietic recovery. Nevertheless, after adjustment for sex, performance score, comorbidity, cytomegalovirus serostatus, and ABO blood group match, cryopreservation was associated with higher 1-year rates of graft failure (HR, 2.26; 95% CI, 1.17 to 4.35; *p* = 0.01) and of 1-year overall mortality (HR, 3.13; 95% CI, 1.60 to 6.11; *p* = 0.0008). The incidence of acute and chronic GVHD were similar. The adjusted probability of 1-year survival were 73% (95% CI, 60% to 84%) in cryopreserved and 91% (95% CI, 86% to 94%) in fresh graft groups. Subset analyses limited to PBSC transplants confirmed higher graft failure and mortality, highlighting a greater effect of cryopreservation on PBCS than in bone marrow grafts [90] These data partly confirmed the observations of Lioznov et al. [87]. Based on this evidence, in patients with severe aplastic anemia, the use of fresh grafts was also recommended during the pandemic, whenever possible. In the same year, using the Center for International Blood and Marrow Transplant Research (CIBMTR) database, Hamadani et al. compared the outcomes of transplants with cryopreserved versus fresh grafts in patients with hematologic malignancies receiving GvHD prophylaxis with post-transplantation cyclophosphamide (ptCY) [91]. A total of 274 adult patients receiving cryo-allografts and 1080 propensity score-matched controls receiving fresh grafts were included in the analysis. In both cohorts, grafts almost completely consisted of PBSCs (93.4%). The two groups showed a similar cumulative incidence of neutrophil and platelet recovery. There was no difference regarding acute GvHD, but in matched-pair regression analysis, cryopreserved grafts were associated with a lower risk of chronic GvHD (HR 0.78; 95% CI, 0.61–0.99%, *p* = 0.04). In the end, the 2-year OS rates were comparable in fresh (60.6%, 95% CI 57.3–63.8%) and cryo-allograft (58.7%, 95% CI 51.9–65.4%) groups [91]. Hsu et al. evaluated the impact of cryopreservation on the outcomes of related and unrelated donor transplants performed from 2013 to 2018 in patients with hematologic malignancies receiving conventional calcineurin inhibitors as GvHD prophylaxis [92]. Comprehensively, 1051 HLA-matched related PBSC, 678 matched unrelated PBSC, and 154 matched related or unrelated bone marrow donors were compared to controls identified by propensity score match (3 to 1) among 5514 patients receiving fresh grafts. Whilst no difference was found in BM transplants, the cryopreservation of related donor PBSC grafts was associated with decreased platelet recovery and increased risk of acute GvHD. Moreover, the cryopreservation of unrelated PBSC grafts was associated with delayed engraftment of neutrophils and platelets and increased risk of NRM and relapse, and decreased progression-free survival (PFS) and OS [92]. In a study involving a smaller series of patients undergoing PBSC transplantation from fully matched sibling donors, Dagdas et al. compared 42 fresh and 30 frozen transplants [93]. The authors reported a delayed neutrophil engraftment in cryopreserved grafts (mean: 14 days vs. 16 days, *p* = 0.006) as well as a slightly higher rate of grade 3–4 liver chronic GVHD in fresh transplants (*p* = 0.046). There was no difference in OS [94]. In the meantime, to better understand which practices could influence hematopoietic recovery in transplants of cryopreserved products, the post-thaw quality of cryopreserved allogeneic products was thoroughly investigated. Purtill et al. evaluated the post-thaw CD34+ cell recovery and viability of 305 allogeneic HPC products cryopreserved at nine laboratories across Australia [55]. The median post-thaw CD34+ cell recovery was 76% (range 6% to 122%) and was significantly influenced by a longer transit time before cryopreservation, total nucleated cell concentration during storage, and types of manipulation before cryopreservation. Similarly, the post-thaw CD34+ cell viability was affected by the length of the pre-cryopreservation transit time and by TNC concentration [55].

It ought to be emphasized that in the above-mentioned studies, the exact reason leading to the cryopreservation of the grafts was in most cases unknown. Conversely, with the progression of COVID-19, the literature data reflected a real-word experience in which graft cryopreservation was accomplished to secure donors and recipients from threats connected to the pandemic: initially, only short-term outcomes were explored, whilst long-term follow-up data have emerged only in most recent studies. The main findings of these studies are summarized in Table 2.
cells-13-00552-t002_Table 2Table 2Impact of cryopreservation in studies including patients transplanted during the COVID-19 pandemic.Authors [Ref]Study TypeDisease/Donor TypesPatients (Graft Source) *Controls (Graft Source) **aGvHD Cumulative Incidence (%)cGvHDCumulative Incidence (%)Overall Survival (%)Relapse Rate (%)Other FindingsMaurer K et al., 2021[94]MulticentricMalignantdiseases/matched related and unrelated and haploidentical donorsCOVID-19 Cohort A 64 32 cryo 32 fresh (14 BM)Pre-COVID-19 cohort B 68 4 cryo 64 fresh(12 BM)Pre-COVID-19 cohort C 764 cryo 72 fresh(21 BM)Day +100Grade II–IV Cohort A 10.9Cohort B 16.2Cohort C 9.2Day 100Grade III–IV Cohort A 6.2Cohort B 1.5Cohort C 1.3NRCohort A 92 Cohort B 94Cohort C 95Day +100 Cohort A 9.4Cohort B 11.8Cohort C 17.1% day +30 WBC chimerism cryo 98 fresh 99 *p* < 0.001% day +30 CD3 chimerism cryo 67fresh 95 *p* = 0.01% day +100 CD3 chimerism cryo 80fresh 97 *p* = 0.03Maurer K et al., 2021[54]Single-center Malignantdiseases/unrelated donors101(PBSCs)203(PBSCs)Day +100Grade II–IV Cryo 17Fresh 9*p* = 0.014Day +100Grade III–IV Cryo 6Fresh 4.5NRDay +100Cryo 96Fresh 96.56 monthsCryo 89Fresh 89Day +100 Cryo 16Fresh 126 months Cryo 22Fresh 20% day +100 NRMCryo 2Fresh 2% 6-month NRM Cryo 2Fresh 4.6More GF if infusion or cryopreservation > 48 hMaurer K et al., 2023[95]Single-center Malignantdiseases/unrelated donors136(PBSCs)251(PBSCs)6 months Grade II–IV Cryo 25Fresh 206-month Grade III–IV Cryo 12Fresh 82 yearsCryo 39Fresh 57*p* < 0.0012 yearsModerate/severe Cryo 18Fresh 31*p* < 0.0012 years Cryo 60Fresh 652 years Cryo 34Fresh 29% 2-year NRMCryo 11Fresh 12Valentini CG et al., 2022[96]Single-center MalignantDiseases, related and unrelated donors32(PBSCs)106(PBSCs)Day +100 Grade II–IVCryo 23.8Fresh 19.5 NRNRNR% 1-year NRMCryo 7.7 Fresh 16.1 Fernandez-Sojo J et al., 2021[98]Single-center Malignant diseases/unrelated donors32(PBSCs)32(PBSCs)Day +100Cryo 41 Fresh 31NRDay +100Cryo 90 Fresh 81 NRDay +100 PFS Cryo 88 Fresh 81Alotaibi A et al., 2021[99]Single-center Malignantdiseases/related, unrelated donors310 (PBSCs)648(PBSCs)Grade II–IV Cryo 49Fresh 50Moderate/severe Cryo 40Fresh 27*p* < 0.0012 yearsCryo 52Fresh 492 yearCryo 23Fresh 18 % 2-year NRMCryo 29 Fresh 36*p* = 0.03In patients without cGVHD, lower relapse incidence in fresh (HR = 0.67, *p* = 0.01)Novitzy-Basso I et al., 2021[100]Single-center Malignantdiseases/related and unrelated donors135(PBSCs)348(PBSCs)Grade II–IV Cryo 47.0Fresh 34.8Moderate/severe Cryo 12.6Fresh 18.42 years Cryo 47.6Fresh 79.4*p* = 0.04In ATG-PTCy cryo 51.9fresh 65.5*p* < 0.052 yearsCryo 28.5Fresh 23.2% 1-year NRM Cryo 20.0Fresh 17.8% 2-year GRFSCryo 41.2Fresh 51.4*p* = 0.04Higher NMR (*p* = 0.005) and lower GRFS in MRD cryo (*p* < 0.001)Guo M et al., 2023[101]Single-center Malignantdiseases/related and unrelated donors34(PBSCs)21(PBSCs)Grade II–IV Cryo 58.8Fresh 42.9Moderate/severeCryo 49.4Fresh 9.5Day +100Cryo 94.1Fresh 100*p* = 0.021 yearCryo 67.6Fresh 90.415 months Cryo 44.1Fresh 28.6% 1-year NRM Cryo 12.8% Fresh 6.3% Facchin G et al., 2022[102]Single-center Malignantdiseases, matched unrelated donors31(PBSCs)23(PBSCs)Grade II–IV aGVHDCryo 56.5Fresh 60.0NR1 year Cryo 80.7Fresh 78.3 NR% 1-year TRM Cryo 13.0Fresh 13.5% 1-year PFS Cryo 71.0 Fresh 65.2 Giammarco S et al., 2023 [103]Single-center Malignantdiseases/related and unrelated donors33(28 PBSCs, 2 BM, 3 CBU)34(17 PBSCs, 14 BM, 3 CBU)NRNRCryo 79 Fresh 82 Cryo 29Fresh 24% GFCryo 6% Fresh 6% Ersal T et al., 2023[104]Single centerMalignantdiseases/full-match sibling donors37(PBSCs)56(PBSCs)Cryo 37.8Fresh 28.6Cryo 10.8Fresh 10.7Day + 100 Cryo 75.7Fresh 96.42 yearCryo 57.3Fresh 67.9NR% Day +100 PFSCryo 94.6Fresh 100% 2-year PFSCryo 82.8Fresh 80.4Keyzner A et al., 2023[105]Single-center Malignantdiseases/related and unrelated donors44(31 PBSCs, 13 BM)37 (27 PBSCs, 10 BM)NRNRNRNRNo impact on chimerismLaroye C et al., 2023[106]Single-center Malignantdiseases/related unrelated57(PBSCs)19(PBSCs)Cryo 54Fresh 20Cryo 79Fresh 10Cryo 8Fresh 5Cryo 12Fresh 10Median CD34+ cell recovery in cryo 69.0%Bankova A et al., 2022 [107]Single-center Malignantdiseases/related and unrelated donors 30(PBSCs)60(PBSCs)NRNRLower in cryo (HR 2.16, 95% CI 1.00–4.67) *p* = 0.050NRHigher NRM in cryo(HR 1.90, 95% CI 0.95–3.79) *p* = 0.071Connelly-Smith L et al., 2023[109]Single-centerMalignantdiseases/related and unrelated donors213(PBSCs)167(PBSCs)Grade II–IVCryo 55.9Fresh 61.71 yearCryo 45Fresh 40Similar OS in multivariate analysis in cryo- and fresh graftsSimilar RR in multivariate analysis in cryo- and fresh graftsSimilar NRM in multivariate analysis in cryo- and fresh graftsDevine SM et al., 2023[110]Multicenter CIBMTR datasetMalignantdiseases/related and unrelated donors 1543 (1361 PBSCs, 182 BM)2499(1834 PBSCs, 665 BM)Day +100Grade II–IVCryo 36.0 Fresh 32.7*p* = 0.0421 yearModerate/severeCryo 16.9 Fresh 19.8 *p* = 0.0231 year Cryo 74.6 Fresh 76.91 year Cryo 22.2 Fresh 19.2%*p* = 0.042% 1-year DFSCryo 63.2Fresh 66.9 % 1-year NRM Cryo 14.7 Fresh 13.9 *p* = 0.027* Patients receiving cryopreserved grafts; ** patients receiving fresh grafts. OS, overall survival; RR, relapse rate; TRM, transplant-related mortality; NRM, non-relapse mortality; PFS, progression-free survival; aGVHD, acute graft-versus-host disease; cGVHD, chronic graft-versus-host disease; PBSCs, peripheral blood stem cells; BM, bone marrow; NR, not reported; ptCy, post-transplant cyclophosphamide; ATG: anti-T lymphocyte globulin; GF, graft failure; RFS, relapse-free survival.


Maurer et al. illustrated the allo-transplant activity data at the Dana-Farber Cancer Institute during the first 3 months of the pandemic. The authors compared outcomes at day +100 of 64 related and unrelated adult transplants with two control groups: 68 transplants performed in the 3 months prior to the pandemic and an additional 76 transplants treated in the 12 months prior to the pandemic [95]. Regarding donor selection, there was a trend for an increase in domestic versus international unrelated donations. Half of the PBSC products collected during pandemic underwent cryopreservation [94]. The cohorts had similar OS, PFS, NRM, early relapse, and acute GVHD incidence. The incidence of graft failure and neutrophil engraftment did not differ across groups, and total leukocyte chimerism and CD3 chimerism at day 30 and day 100 were similar. Nevertheless, cryopreserved products were associated with a decreased total leukocyte and CD3 chimerism at both day 30 and day 100 in comparison with fresh allografts [94]. The same group subsequently confirmed these data in a larger series of patients transplanted with unrelated donor grafts from January 2019 to December 2020: 101 received cryopreserved and 203 fresh PBSC grafts [54]. Although short-term outcomes were similar in recipients of fresh or cryopreserved grafts, there was a trend toward increased graft failure, particularly for grafts with overall product age ≥ 48 h [54]. Moreover, the recipients of PBSC cryopreserved grafts exhibited an impaired immune reconstitution. Peripheral blood CD3+ chimerism was significantly impaired at day 30 and 100 after transplant, with 34% of cryo-graft recipients having CD3+ chimerism ≤ 50%, compared with 14% patients who received fresh PBSCs. In addition, there was a delayed platelet engraftment in the cryopreserved cohort compared with controls, as well as an increase in the incidence of acute grade II to IV GVHD [54]. Overall, the authors suggested caution in utilizing cryopreserved allografts, particularly those requiring longer times in transit holding. In fact, although early clinical outcomes did not appear to be compromised, low T cell chimerism and impaired immune reconstitution could affect longer-term outcomes [54]. These results were further confirmed in a subsequent report involving a larger population size and an extended observation period [95]. Additional information on longer-term outcomes was provided, showing that cryopreserved and fresh graft transplants had a similar OS and relapse rate, but cryopreserved graft recipients had a lower incidence of chronic GvHD, either any grade or II–IV grade [95].

Valentini et al. compared data related to allogeneic related and unrelated PBSC transplants performed at an Italian transplant center from 2018 [96]. Patients were grouped according to the COVID-19 period (167 before and 45 during the pandemic). During the pandemic, all PBSC products were cryopreserved, with a median storage time of 20 days. There was no difference in engraftment, acute GvHD and NRM between cryopreserved and fresh products [96]. Mfarrej et al. analyzed post-thaw cell recovery in 42 allogeneic PBSC grafts (28 from related and 14 from unrelated donors), cryopreserved between March and July 2020 at a transplant center in France [97]. The authors reported a lower recovery of viable CD34+cells in unrelated donor grafts, which correlated with the time from collection to cryopreservation. All patients achieved neutrophil engraftment. Nevertheless, 2 out of 42 grafts were not transplanted [97]. At Vall d’ Hebron Barcelona Hospital, Fernandez-Sojo et al. compared 32 patients receiving unrelated donor cryopreserved PBSC transplant during COVID-19 pandemic with 32 patients receiving fresh PBSC allografts in the immediately previous period [98]. Groups were comparable for age, gender, hematologic malignancy, donor/recipient ABO, sex, HLA, CMV serostatus compatibility, conditioning, GvHD prophylaxis, and prognosis index. The two cohorts showed no differences in neutrophil and platelet engraftment, full donor chimerism, acute GvHD, preemptive therapy on CMV disease, PFS, and OS. At the time of writing, the authors reported that 6 of 47 (13%) donated and cryopreserved allografts were not transplanted due to patient worsening or refusal [98]. A concern regarding the non-used allografts was also reported by Purtill et al. [112]. The authors evaluated 191 PBSC products worldwide collected between April and September 2021 for Australian transplant center patients. Among these products, 22 (12%) were still not infused with a minimum follow-up of 9 months. Causes were mostly related to disease progression or interval illness. In comparison, only 3 out of 339 products (0.8%) were not infused in 2019 [112]. Two sized studies described the experience of the Princess Margaret Cancer Centre in Canada. First, Alotaibi et al. evaluated the effect of graft cryopreservation in 958 related, unrelated and haploidentical donor transplants followed between January 2010 and October 2018: 648 patients received fresh grafts and 310 cryopreserved grafts [99]. The authors did not observe differences between fresh and cryopreserved grafts for neutrophil and platelet engraftment, rate of graft failure, and grade II-IV acute GvHD, whereas moderate/severe chronic GvHD was observed in 176 (27%) cryopreserved and 123 (40%) fresh graft recipients. At multivariable analysis, cryopreservation had no effect on OS and relapse incidence, while fresh grafts demonstrated borderline increased NRM. Moreover, in patients without chronic GvHD, relapse incidence was significantly lower in fresh than cryo-graft transplants (HR = 0.67, *p* = 0.01). The authors concluded that cryopreservation is a safe option for allogeneic HCT. Of note, in this large series, only one third of patients received anti-T-lymphocyte globulin (ATG) and PTCy as GvHD prophylaxis [99]. Subsequently, Novitzky-Basso et al. analyzed the outcomes of a smaller series of 483 patients transplanted at the same hospital from August 2017 to August 2020: 348 received fresh grafts and 135 cryopreserved grafts [100]. Patients receiving cryo-allografts had a reduced survival and GVHD- and relapse-free survival, reduced incidence of chronic GvHD, delay in neutrophil engraftment, and higher graft failure, with no significant difference in relapse incidence or acute GvHD. The detrimental effect of cryopreservation was almost exclusively confined to recipients of cryopreserved matched-related donor grafts, who showed significantly worse OS, NRM, GvHD, and relapse-free survival (GRFS) compared with fresh grafts. A multivariable analysis of the related and unrelated entire cohort showed a significant impact of cryopreservation on OS, relapse, chronic GvHD, graft failure, and GRFS. At a subset analysis, according to the GvHD prophylaxis, patients in the cryopreserved graft group receiving ATG and PTCy had a 2-year survival probability of 51.9% in comparison to 65.5% of the fresh graft group [100]. Therefore, the authors concluded that the inferior outcomes of cryopreserved transplants were possibly due to the combination of ATG and PTCy impacting the differential tolerance to cryopreservation of various components of the grafts [100]. Similar findings were reported by Guo et al. in a single center study involving 34 patients receiving cryopreserved PBSC transplant and 31 controls [101]. Conversely, various single-institution studies found no differences between cryo- and fresh grafts in unrelated donor transplants [102,103,104,105,106]. A further single institution study reported delayed engraftment and a trend toward poorer immuno-reconstitution of cryopreserved allografts overall in myeloablative conditioning transplant [107]. In a retrospective study carried out in Japan, 235 cryopreserved unrelated BM and 118 PBSC transplants performed during the pandemic period were compared with a large pre-pandemic control cohort including 4133 BM and 720 PBSC transplants [108]. In the multivariate analysis, cryopreservation in PBSCTs significantly delayed neutrophil and platelet engraftment, whereas it did not affect neutrophil engraftment in BM transplants [108]. An observational study from the Seattle group compared cryopreserved grafts (n = 213) with a historical cohort receiving fresh grafts (n = 167): basically, a slight delay in platelet engraftment for cryopreserved grafts was confirmed [109]. In addition, cryo-allografts were associated with a greater proportion of patients with lower CD3 cell chimerism at day +28 [109]. Nevertheless, in multivariable analyses, fresh and cryopreserved grafts had a similar overall mortality, NRM, and relapse rate. There was no demonstrable difference in the risk of chronic GvHD. The authors concluded that PBSC cryopreservation is a reasonable and safe option [109]. Finally, using the CIBMTR database, Devine et al. reported the impact of cryopreservation on OS and other outcomes at 1 year after transplant [110]. A total of 1543 recipients of cryopreserved allografts transplanted at US centers during the first 6 months of the pandemic were compared with 2499 recipients of fresh allografts transplanted during a 6-month period in 2019. More than 70% of patients received PBSCs, with a minority receiving BM grafts. In the univariate analysis, patients receiving cryopreserved PBSC transplants showed delayed neutrophil and platelet engraftment, lower incidence of moderate/severe chronic GvHD, and reduced DFS. In the multivariable regression analysis, there was no difference in the 1-year OS between the groups. However, the 1-year DFS was lower in the group with cryopreserved grafts (HR, 1.18; 95% CI, 1.05–1.33; *p* = 0.006), with no difference in the 1-year NRM. The risk of primary graft failure was higher in the cryopreserved group, and the probability of platelet recovery by day 100+ was lower. Finally, patients receiving cryopreserved grafts had a higher risk of moderate-to-severe chronic GvHD and relapse at 1 year than those in the group with fresh grafts (HR, 1.21; 95% CI, 1.04–1.41; *p* = 0.01) [110].

## 5. Overview on the Evidence Provided by Clinical Studies

Considering all the above-mentioned studies from a comprehensive perspective, the evidence provided can appear sometimes conflicting. Table 3 provides a global overview of cryopreservation effects on engraftment, incidence of acute and chronic GVHD, overall survival, disease-free survival, and non-relapse mortality. 

The evidence that cryopreservation can affect the hematopoietic recovery is reported by several authors, including most recent sizeable studies [54,88,91,92,93,100,101,104,107,108,109,110]. Despite this effect being modest in most cases (Figure 2), some studies (including those performed in the pre-COVID-19 period) clearly reported an increased graft failure rate with cryopreserved products, namely if using PBSC grafts [54,87,90]. This risk is higher for products with a longer transit time prior to cryopreservation or infusion [54]. Practically speaking, cryopreserving products at the collection site, soon after the harvest, could mitigate this hazard.

The increased incidence of graft failure might be in part attributed to an immune-mediated mechanism. Overall, this effect can underlie the influence of freezing on several other transplant outcomes. In this regard, it should be emphasized that cryopreservation can select different immune cell populations and influence their function. The possibility of developing GvHD is therefore modulated by the interaction of lingering cells with the administered pharmacological prophylaxis. This interplay might explain the rather contradictory data reported in GvHD setting. Regarding aGvHD, the effect exerted by cryopreservation on the T-reg subset could theoretically result in an increased risk. However, most authors did not observe a significant impact of cryopreservation on aGvHD; two studies reported a reduced incidence exclusively in cryopreserved BM [85] and related PBSCs [92], whilst Parody et al. [89] and Maurer et al. [54] found a reduction in the aGVHD incidence. Notably, in studies of Parody et al. [89] and Maurer et al. [54], no or a few patients received PTCy, respectively, a drug which prevents aGvHD by favoring the selection of the T-reg subset [54]. On the contrary, in the setting of cGvHD, several studies found that severity and incidence may be reduced with cryopreserved grafts even in patients receiving PTCy [91,100,110], whereas this effect has also been observed when GvHD prophylaxis is carried out with other agents [93,95]. In the only study reporting an increased incidence of cGvHD with cryo-grafts, this observation could have been biased by the higher rate of in vivo T cell depletion in the fresh graft group [99]. Besides the impact on GvHD, cryopreservation can also impair the graft-versus-leukemia response, as evidenced by the higher relapse rate and lower DFS emerging in most recent and sizeable studies [91,107,110]. Moreover, some authors also reported in cryo-transplants a significant increase in NRM [92,100,107] or a trend for a higher NRM not further confirmed in a multivariate analysis [110]. It is likely that the impaired immune reconstitution could account for this effect [54]. Finally, a group of studies heterogeneous for period, sample size, and patient populations suggest a cumulative detrimental effect of cryopreservation on OS [90,92,100,101,107]. Nevertheless, further studies with a longer follow-up and larger sample size are necessary to confirm this finding. 

## 6. Conclusions

According to a recent survey by the Cellular Therapy and Immunobiology Working Party of the EBMT group, after the pandemic, the number of transplant centers cryopreserving HSC products has risen from 7.9% to 90% [113]. During the pandemic period, freezing HSC grafts has been revealed as advantageous from the perspective of both donors and recipients, offering higher flexibility in scheduling HSC collection, allowing for a delay in the transplant depending on recipient clinical needs, and minimizing the impact of an eventual donor SARS-CoV2 infection [114,115]. Moreover, planning donations according to the donor requirements could probably increase the likelihood of a product’s availability. In contrast, however, cryopreservation implies added costs connected to the procedure itself (the burden of higher work hours for the laboratory staff), the use of consumables and disposables, and the microbial and viability tests necessary for the product release. Lastly, ethical concerns exist relating to the fact that harvested and cryopreserved products might not get transplanted due to circumstances related to recipients (i.e., sudden worsening or death).

When cryopreserving HSC grafts was urgently recommended at the onset of the COVID-19 pandemic, there was an awareness that cryopreservation causes a loss of CD34+ cells and that long transit before freezing or high total nucleated cell concentration at freezing could make the loss more pronounced. Nevertheless, most studies displayed no cryopreservation effect on engraftment and hematopoietic recovery. There was also biological evidence that different cell types comprising the graft exhibit different levels of sensitivity to cryopreservation and storage. In recent years, the knowledge in this field has expanded incredibly, showing that cryopreservation itself, as well as the different variables associated with cryopreservation, may differently affect not only the viability but also the function of various graft components (Figure 3).

Undeniably, cryopreserving allogeneic grafts has several logistic advantages over transplants of fresh products, and offered a lifesaving opportunity to patients whose allogeneic transplants could not be postponed until the end of the COVID-19 pandemic. However, this achievement has come at the price of an increase in the cost and complexity of transplant organization, and it has raised concerns regarding efficacy, safety, and ethics. In the context of a return to normality and faced with longer follow-up data suggesting poorer long-term outcomes compared with transplants with fresh products, the universal adoption of this strategy might no longer be justified. Since there is no consensus on the benefits and drawbacks, the existing data and specific local factors should be weighed by each transplant center and in each individual case before the decision is made. 

## Figures and Tables

**Figure 1 cells-13-00552-f001:**
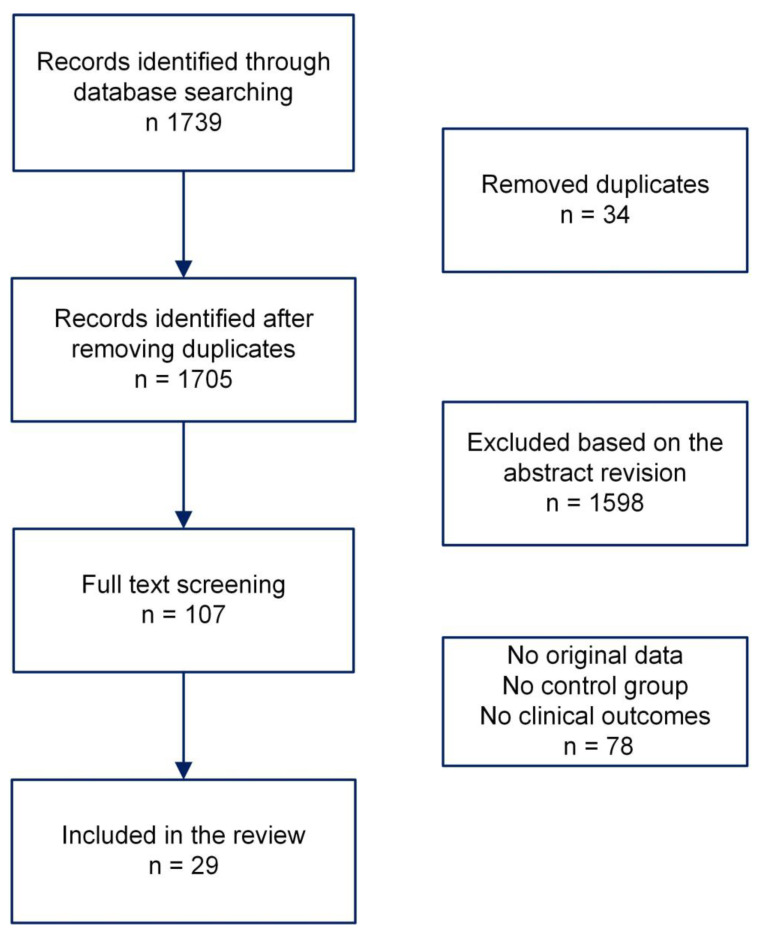
Flow chart of the literature revision.

**Figure 2 cells-13-00552-f002:**
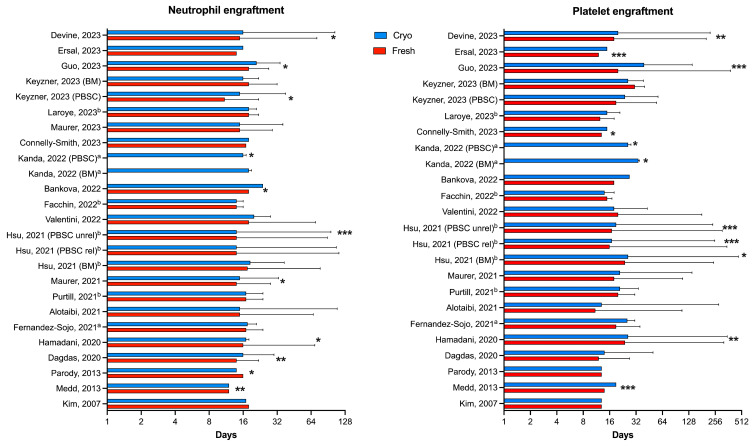
**Days to neutrophil and platelet recovery** [86,88,89,91,92,93,94,95,96,98,99,101,102,104,105,106,107,108,109,110,111]. Data are given as median and range, median and 95% confidence interval ^a^, or median and interquartile range ^b^. Asterisks indicate significant difference between cryopreserved and fresh grafts (* *p* ≤ 0.05, ** *p* ≤ 0.01, and *** *p* ≤ 0.001). The study of Kanda [108] reported no median recovery in fresh grafts, but in multivariate analysis, cryo-grafts were associated with a significantly delayed engraftment of neutrophil in PBSCs and of platelet in BM and PBSC. The study of Maurer (2023) [95] reported no data on platelet engraftment.

**Figure 3 cells-13-00552-f003:**
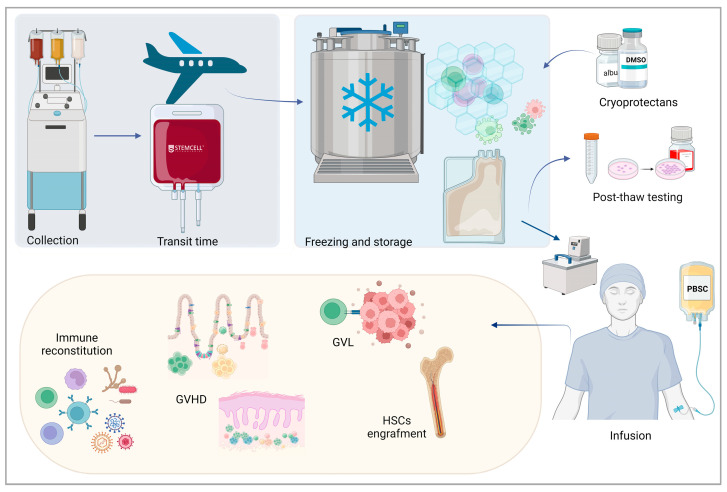
**Variables affecting transplant outcomes in transplant of cryopreserved grafts.** Hematopoietic stem cell (HSC) graft is collected by apheresis from the peripheral blood of a G-CSF mobilized donor. A significant amount of time may elapse between donation and cryopreservation: the “transit time” spans between the end of collection at the donation site and the delivery to the receipt center. After the addition of a solution containing the dimethyl sulfoxide (DMSO) cryoprotectant, the product undergoes controlled-rate freezing and it is subsequently stored in liquid or vapor nitrogen phase. Cryopreservation can differentially select for cellular sub-populations which may be more tolerant to the molecular and physical stresses experienced during the process. Cryopreserved HSCs are thawed using dry warming or a 37 °C water bath and then infused into the recipient through a venous access. Post-thawing tests, encompassing assessment of membrane integrity, cell function, and biochemical alterations, are used to analyze the cell quality pre-release. Graft composition after thawing contributes to hematopoietic engraftment and immune reconstitution and modulates the graft-versus-leukemia (GVL) effect and the risk of developing acute and chronic graft-versus-host disease (GVHD).

**Table 3 cells-13-00552-t003:** Impact of cryopreservation on clinical transplant outcomes.

Authors [Ref], Graft Source	Year	Engraftment		aGvHD	cGvHD	OS	DFS/PFS	NRM	Other Findings
		N	PLT						
Eckardt [85], BM	1993	similar	similar	lower	NR	NR	NR/NR	NR	
Kim [86], PBSCs	2007	similar	similar	similar	similar	similar	NR/similar	similar	similar lymphocyte recovery
Lioznov [87], BM	2008	similar	similar	NR	NR	NR	NR/NR	NR	
Lioznov [87], PBSCs	2008	more graft failure		NR	NR	NR	NR/NR	NR	
Medd [88], PBSCs	2013	delayed	delayed	similar	similar	similar	NR/similar	NR	
Parody [89], PBSCs	2013	faster	similar	higher	similar	similar	NR/NR	similar	
Eapen [90], PBSCs/BM	2020	more graft failures in PBSCs		similar	similar	lower in PBSCs	NR/NR	NR	
Hamadani [91], PBSCs/BM	2020	delayed	delayed	similar	lower	similar	lower/similar	similar	
Hsu [92], PBSCs unrelated	2021	delayed	delayed	similar	NR	lower	NR/similar	higher	
Hsu [92], PBSCs related	2021	similar	delayed	lower	NR	similar	NR/similar	similar	
Hsu [92], BM	2021	similar	delayed	similar	NR	similar	NR/similar	similar	
Dagdas [93], PBSCs	2020	delayed	similar	similar	lower (liver)	similar	NR/NR	similar	similar relapse rate
Maurer [94], PBSCs/BM	2021	similar	similar	similar	NR	similar	NR/similar	similar	
Maurer [54], PBSCs	2021	delayed	delayed	higher	NR	similar	NR/NR	similar	lower lymphocyte recovery and CD3 chimerism
Maurer [95], PBSCs	2023	NR	NR	similar	lower	similar	similar/similar	similar	similar 1 y CD3 chimerism
Valentini [96], PBSCs	2022	similar	similar	similar	NR	NR	NR/NR	similar	similar relapse rate
Fernandez-Sojo [98], PBSCs	2021	similar	similar	similar	NR	similar	NR/similar	NR	
Alotaibi [99], PBSCs	2021	similar	similar	similar	higher	similar	NR/NR		similar relapse rate
Novitzy-Basso [100], PBSCs	2021	delayed	similar	similar	lower	lower	NR/NR	higher	GRFS lower
Guo [101], PBSCs	2023	delayed	delayed	similar	similar	lower	similar/NR	similar	
Facchin [102], PBSCs	2022	similar	similar	similar	NR	similar	NR/similar	NR	
Giammarco [103], PBSCs/BM, CB	2023	similar graft failure		NR	NR	similar	NR/NR	NR	
Ersal [104], PBSCs	2023	similar	delayed	similar	similar	similar	NR/similar	similar	similar 30 d chimerism
Keyzner [105], PBSCs/BM	2023	similar	similar	NR	NR	NR	NR/NR	NR	similar 30 d/100 d chimerism
Laroye [106], PBSCs	2023	similar	similar	similar	similar	similar	similar	similar	
Bankova [107], PBSCs	2022	delayed in RIC	delayed in RIC	NR	NR	lower	lower/NR	higher	
Kanda [108], PBSCs	2022	delayed	delayed	NR	NR	NR	NR/NR	NR	
Kanda [108], BM	202	similar	delayed	NR	NR	NR	NR/NR	NR	
Connelly-Smith [109], PBSCs	2023	similar	delayed	similar	similar	similar	NR/similar	similar	
Devine [110], PBSCs/BM	2023	delayed	delayed	similar	lower	similar	lower/NR	similar	higher relapse rate

PBSCs, peripheral blood stem cells; BM, bone marrow; CBU, cord blood unit; aGVHD, acute graft-versus-host disease; cGVHD, chronic graft-versus-host disease; OS, overall survival; DFS, disease-free survival; NRM, non-relapse mortality; RIC, reduced intensity conditioning regimen; NR, not reported.

## Data Availability

No new data were created or analyzed in this study. Data sharing is not applicable to this article.

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
