# Peer review of "Pros and Cons of Cryopreserving Allogeneic Stem Cell Products"

_cells, 2024, doi:10.3390/cells13060552_

Round 1

Reviewer 1 Report

Comments and Suggestions for Authors

Comments for Valentini et al.

The review by Valentini et al. titled “Pros and cons of cryopreserving allogeneic peripheral stem cell products” summarizes literature regarding the long-term transplant outcome from patients who received cryopreserved allogeneic hematopoietic stem cell products.

Major comments,

1. Topic described in this review has been published by other groups in the past which decreases its novelty.

2. Please explain the rationale for the title “Pros and cons of cryopreserving allogeneic peripheral stem cell products” when the review contains summary of findings from patients who received bone marrow products. For example, in page 11 of 30, the Keymer et al contained findings from BM recipients.

3. Table 1 “Main findings of studies included in the review” is extremely confusing and overwhelms the reader with loads of information.

I suggest breaking down the information into several parts. For example:

a. Swimmer plot showing Cryo-allo products (days post-transplant) on the x-axis and the neutrophil and platelet recovery on the y axis from relevant papers.

b. Break down the analysis into 30 day and 100-day post-transplant (allo-Apheresis vs allo-BM) and the effects on:

·         % of CD34+, NK, T-memory cells

·         platelet counts

·         neutrophil counts

·         GvHD

·         VOD

·         Chimerism

·         Engraftment

c. Discusses disease severity of these patients and how it impacts point “b” above.

Author Response

The review by Valentini et al. titled “Pros and cons of cryopreserving allogeneic peripheral stem cell products” summarizes literature regarding the long-term transplant outcome from patients who received cryopreserved allogeneic hematopoietic stem cell products.

Major comments

  1. Topic described in this review has been published by other groups in the past which decreases its novelty.

Reply. We agree with Reviewer that other groups have published similar articles in the past. We included in this revision most recent studies addressing the impact of cryopreserved grafts in a longer follow up period. Although our article cannot provide original data, we would like to humbly contribute an updated portrait of current literature for readers.

  1. Please explain the rationale for the title “Pros and cons of cryopreserving allogeneic peripheral stem cell products” when the review contains summary of findings from patients who received bone marrow products. For example, in page 11 of 30, the Keymer et al contained findings from BM recipients.

Reply. We thank the Reviewer for underlining this issue. Although most part of studies included PBSC transplants, the issue of cryopreservation has also been investigated when using bone marrow as a graft source. We therefore removed the term peripheral from the title, and better addressed the differences observed between peripheral stem cell products and bone marrow in studies, including both types of grafts.   

  1. Table 1 “Main findings of studies included in the review” is extremely confusing and overwhelms the reader with loads of information.

I suggest breaking down the information into several parts. For example:

  1. Swimmer plot showing Cryo-allo products (days post-transplant) on the x-axis and the neutrophil and platelet recovery on the y axis from relevant papers.
  2. Break down the analysis into 30 day and 100-day post-transplant (allo-Apheresis vs allo-BM) and the effects on:
  • % of CD34+, NK, T-memory cells
  • platelet counts
  • neutrophil counts
  • GvHD
  • VOD
  • Chimerism
  • Engraftment
  1. Discusses disease severity of these patients and how it impacts point “b” above.

Reply. We thank the Reviewer for these constructive suggestions. We reorganized the manuscript breaking down the information into four different parts. 

  1. We use a Swimmer plot-like graph to show the effect of cryopreservation on hematopoietic recovery. The median number of days to neutrophil and platelet engraftment reported in various studies is provided: cryopreserved and fresh grafts are compared, and significant differences are indicated. (new Figure 2).
  2. We emphasized that studies performed before the COVID-19 pandemic did not report the reasons for cryopreserving grafts and this could represent a bias for the correct interpretation of the eventual negative effects of cryopreservation on transplant outcomes. Therefore, we subdivided the list of revised studies into two main periods and summarized them in two distinct Tables: new Table 1 includes studies carried out before the COVID-19 pandemic, while new Table 2 includes those carried out during the pandemic when graft cryopreservation was officially recommended. Data on chimerism and immune recovery have been included among other transplant outcomes.
  3. Finally,  we introduced a new paragraph "Overview of evidence provided by clinical studies" and a a new Table 3 to compare and discuss different results on clinical outcomes (engraftment, incidence of acute and chronic GVHD, overall survival, disease-free survival, and non-relapse mortality).

Reviewer 2 Report

Comments and Suggestions for Authors

The review paper titled "Pros and Cons of Cryopreserving Allogeneic Peripheral Stem Cell Products" provides valuable insights into the evolving practice of transplanting cryopreserved allografts, particularly in the context of the COVID-19 pandemic. The paper delves into the basic principles of cell cryopreservation and examines the effects of cryopreservation on different graft components. Additionally, it conducts a literature review comparing transplant outcomes in patients receiving cryopreserved and fresh grafts.

  1. Importance and Interest: The manuscript addresses a timely and important topic, given the widespread adoption of cryopreservation during the COVID-19 pandemic.

The first part of the paper effectively covers the basic principles of cryopreservation. However, there is room for improvement in the detailed explanation of further cryopreservation methods except for DMSO e.g: vitrification, which could be made clearer and more informative to enhance reader understanding and knowledge.

  1. Cryopreservation of HSC Grafts: The second part of the paper focuses on the cryopreservation of hematopoietic stem cell (HSC) grafts. To enhance clarity, it would be beneficial to provide a succinct bottom line summarizing official recommendations for preservation, particularly for different products such as peripheral stem cell transplant (PSCT), bone marrow (BM), donor lymphocyte infusion (DLI), and chimeric antigen receptor T-cell (CAR-T) therapy.
  2. Clinical Impact of Allogeneic PST Cryopreservation: The third part of the paper examines the clinical impact of cryopreservation on allogeneic peripheral stem cell transplant (PST). In this section, to improve the presentation of data, the authors should consider amending the table. Specifically, studies sourced from databases such as the Center for International Blood and Marrow Transplant Research (CIBMTR) or the Japan Bone Marrow Transplant Registry Program should be clearly identified as multicenter studies. Additionally, uniformity in reporting key metrics, such as engraftment percentages (Instead of days with P values that are not explained) and overall survival rates (the different metrics should be highlighted), would enhance comparability across studies.

The description of the manuscripts stated in the table should be revised. Instead of describing several studies, subsections should be created referring to the different results in specific issues shown in the table, such as engraftment, graft-versus-host disease (GVHD), and overall survival (OS). The focus should be on highlighting contradictory or non-contradictory results between studies, with possible explanations for the observed differences between studies, considering for example differences in study populations and transplantation practices.

4.     Conclusion: The conclusion should refer to the data presented and current guidelines regarding cryopreserved or non-cryopreserved grafts. It should discuss how the presented data aligns with current practices and guidelines, providing insights into the implications for clinical decision-making in the transplantation setting.

Author Response

The review paper titled "Pros and Cons of Cryopreserving Allogeneic Peripheral Stem Cell Products" provides valuable insights into the evolving practice of transplanting cryopreserved allografts, particularly in the context of the COVID-19 pandemic. The paper delves into the basic principles of cell cryopreservation and examines the effects of cryopreservation on different graft components. Additionally, it conducts a literature review comparing transplant outcomes in patients receiving cryopreserved and fresh grafts.

  1. Importance and Interest: The manuscript addresses a timely and important topic, given the widespread adoption of cryopreservation during the COVID-19 pandemic.

The first part of the paper effectively covers the basic principles of cryopreservation. However, there is room for improvement in the detailed explanation of further cryopreservation methods except for DMSO e.g.: vitrification, which could be made clearer and more informative to enhance reader understanding and knowledge.

Reply: We thank the reviewer for this stimulating comment. As suggested, the vitrification process has been now widely discussed, to provide readers with more complete and clear information. Moreover, we introduced and discussed two additional innovative approaches to cryopreservation, super flash freezing and nanowarming (new paragraph 2, lines 65-106 of the revised manuscript).

  1. Cryopreservation of HSC Grafts: The second part of the paper focuses on the cryopreservation of hematopoietic stem cell (HSC) grafts. To enhance clarity, it would be beneficial to provide a succinct bottom line summarizing official recommendations for preservation, particularly for different products such as peripheral stem cell transplant (PSCT), bone marrow (BM), donor lymphocyte infusion (DLI), and chimeric antigen receptor T-cell (CAR-T) therapy.

Reply: We thank the reviewer for this constructive suggestion. We added to the revised manuscript a brief paragraph summarizing the recommendations for different cell therapy products, including CAR-T cells and donor lymphocyte infusion (lines 241-257 of the revised manuscript).

  1. Clinical Impact of Allogeneic PST Cryopreservation: The third part of the paper examines the clinical impact of cryopreservation on allogeneic peripheral stem cell transplant (PST). In this section, to improve the presentation of data, the authors should consider amending the table. Specifically, studies sourced from databases such as the Center for International Blood and Marrow Transplant Research (CIBMTR) or the Japan Bone Marrow Transplant Registry Program should be clearly identified as multicenter studies. Additionally, uniformity in reporting key metrics, such as engraftment percentages (Instead of days with P values that are not explained) and overall survival rates (the different metrics should be highlighted), would enhance comparability across studies.

Reply: We thank the Reviewer for these constructive suggestions. We rearranged the manuscript breaking down the information into four different parts, amending figures and tables as suggested. First, we added a Swimmer plot-like graph to show the effect of cryopreservation on the hematopoietic recovery (new Figure 2). We then subdivided the list of revised studies in two main periods and summarized them in two distinct Tables: the new Table 1 includes studies carried out before the COVID-19 pandemic, while the new Table 2 includes those carried out during the pandemic when graft cryopreservation was officially recommended. Whenever present, data on chimerism and immune recovery have been included among other transplant outcomes. The design of the study (single center vs multicenter) was corrected. Finally, as suggested, we condensed the impact of cryopreservation on clinical transplant outcomes in a new Table 3, summarizing findings from all reviewed studies to enhance comparability across studies.

  1. The description of the manuscripts stated in the table should be revised. Instead of describing several studies, subsections should be created referring to the different results in specific issues shown in the table, such as engraftment, graft-versus-host disease (GVHD), and overall survival (OS). The focus should be on highlighting contradictory or non-contradictory results between studies, with possible explanations for the observed differences between studies, considering for example differences in study populations and transplantation practices.

Reply: we thank the reviewer for this constructive suggestion. We adopted a chronological approach to introduce the studies throughout the discussion. The rationale for this choice resides in the extreme heterogeneity of the manuscripts presented: studies performed before the COVID-19 pandemic did not report the reasons for cryopreserving grafts and this poses a possible selection bias for the correct interpretation of the eventual negative effects of cryopreservation on transplant outcomes. Moreover, studies performed in the early period of the COVID-19 era are understandably characterized by limited follow-ups and tend to focus on short-term outcomes. At complement, we reorganized the table and figured to stress more clearly the different results provided for specific issues. Specifically, we add a swimmer plot-like graph to show the effect of cryopreservation on hematopoietic recovery, indicating statistically significant differences (New Figure 2). New Table 1 includes studies carried out before the COVID-19 pandemic and new Table 2 those carried out during the COVID-49 pandemic, when cryopreservation was officially recommended. For both tables, we harmonized the reporting key metrics, highlighting differences in the incidence of acute and chronic GVHD, relapse, and overall survival, providing the readers with a more direct comparison. As suggested, we specified which studies were sourced from registries (i.e. CIBMTR). Finally, we created a new paragraph (Overview on the evidence provided by clinical studies) and a new Table 3 to compare and discuss different results on clinical outcomes (engraftment, incidence of acute and chronic GVHD, overall survival, disease-free survival, and non-relapse mortality).

  1. Conclusion: The conclusion should refer to the data presented and current guidelines regarding cryopreserved or non-cryopreserved grafts. It should discuss how the presented data aligns with current practices and guidelines, providing insights into the implications for clinical decision-making in the transplantation setting.

Reply: we thank the reviewer for these observations. Accordingly, we expanded the conclusion section, discussing how clinical practice should be guided based on current evidence.

Round 2

Reviewer 1 Report

Comments and Suggestions for Authors

The authors have sufficiently address the issues in the manuscript.

I would consider it acceptable in present form.

Author Response

We thank the Reviewer for the positive comments. 

Reviewer 2 Report

Comments and Suggestions for Authors

Thank you for revising the manuscript on "Pros and cons of cryopreserving allogeneic stem cell products." I appreciate the efforts made to address my suggestions.

I recommend reconsidering the sentence added on page 22, row 674, which states, "Indeed, cryopreservation should be opted in selected cases when the benefits clearly outweigh the predicted drawbacks." Since there's no consensus on the benefits and drawbacks, I suggest emphasizing that each transplant center and individual case should weigh the existing data and specific local factors before deciding.

Comments on the Quality of English Language

I suggest to revise some of the new paragraphs for clarity (page 7 and 21).

I noticed a few English errors and typos:

page 2

act have" should be "act, having" f (line 97)

page 5

"leukapheresis material" should be clarified as "leukapheresis products" (line 246).

"escalated dose schedule at increasing the cell dosages" should be simplified to "gradually increasing the cell dosages" for clarity (line 256).

page 19

provide" should be "provides" in "Table 3 provide a global overview..." (line 588)

"can underlies" should be "can underlie" (line 599)

  1. "a few of patients" should be "a few patients" (line 608).
  2. "could have be" should be "could have been" (line 615).
  3. "observed observed" should be corrected to "observed" (line 619).

Author Response

We deeply thank the Reviewer for all the comments and suggestions provided in revising our manuscript.

We carefully revised the text and amended all the indicated typos (we apologize for the inaccuracy). Moreover, as suggested, we modified the last sentence of the Conclusion paragraph, emphasizing that, in the absence of definite evidence, each transplant center should weigh the pros and cons of cryopreserving grafts based on individual cases and specific factors. 

We highlighted all changes performed throughout the text.